# The Anti-Inflammatory Effect of Low Molecular Weight Fucoidan from *Sargassum siliquastrum* in Lipopolysaccharide-Stimulated RAW 264.7 Macrophages via Inhibiting NF-κB/MAPK Signaling Pathways

**DOI:** 10.3390/md21060347

**Published:** 2023-06-04

**Authors:** Arachchige Maheshika Kumari Jayasinghe, Kirinde Gedara Isuru Sandanuwan Kirindage, Ilekuttige Priyan Shanura Fernando, Kil-Nam Kim, Jae-Young Oh, Ginnae Ahn

**Affiliations:** 1Department of Food Technology and Nutrition, Chonnam National University, Yeosu 59626, Republic of Korea; 218385@jnu.ac.kr (A.M.K.J.); 218388@jnu.ac.kr (K.G.I.S.K.); 2Department of Agricultural, Food and Nutritional Science, University of Alberta, 4-10 Ag/For Building, Edmonton, AB T6G 2P5, Canada; ilekutti@ualberta.ca; 3Chuncheon Center, Korea Basic Science Institute (KBSI), Chuncheon 24341, Republic of Korea; knkim@kbsi.re.kr; 4Food Safety and Processing Research Division, National Institute of Fisheries Science, Busan 46083, Republic of Korea; ojy0724@naver.com; 5Department of Marine Bio-Food Sciences, Chonnam National University, Yeosu 59626, Republic of Korea

**Keywords:** *Sargassum siliquastrum*, low molecular weight fucoidan, RAW 264.7 macrophages, anti-inflammation, NF-κB/MAPK signaling

## Abstract

Brown seaweed is a rich source of fucoidan, which exhibits a variety of biological activities. The present study discloses the protective effect of low molecular weight fucoidan (FSSQ) isolated from an edible brown alga, *Sargassum siliquastrum,* on lipopolysaccharide (LPS)-stimulated inflammatory responses in RAW 264.7 macrophages. The findings of the study revealed that FSSQ increases cell viability while decreasing intracellular reactive oxygen species production in LPS-stimulated RAW 264.7 macrophages dose-dependently. FSSQ reduced the iNOS and COX-2 expression, inhibiting the NO and prostaglandin E_2_ production. Furthermore, mRNA expression of IL-1β, IL-6, and TNF-α was downregulated by FSSQ via modulating MAPK and NF-κB signaling. The NLRP3 inflammasome protein complex, including NLRP3, ASC, and caspase-1, as well as the subsequent release of pro-inflammatory cytokines, such as IL-1β and IL-18, release in LPS-stimulated RAW 264.7 macrophages was inhibited by FSSQ. The cytoprotective effect of FSSQ is indicated via Nrf2/HO-1 signaling activation, which is considerably reduced upon suppression of HO-1 activity by ZnPP. Collectively, the study revealed the therapeutic potential of FSSQ against inflammatory responses in LPS-stimulated RAW 264.7 macrophages. Moreover, the study suggests further investigations on commercially viable methods for fucoidan isolation.

## 1. Introduction

Marine resources have drawn interest in the field of research focused on bioactive metabolites that could provide nutritious functional foods, cosmeceuticals, and promising therapeutic agents against disease conditions. Seaweed, also known as marine macroalgae, are rich bioresources found worldwide. Marine macroalgae are classified into three main groups based on their pigment content: brown algae, red algae, and green algae. Among them, brown algae species are popular in research due to their abundance of secondary bioactive metabolites, including sulfated polysaccharides, phlorotannins, fucoxanthin, sterols, and other phenolic compounds with a variety of biological properties [1]. In addition, brown algae have been used for centuries in many countries, especially in East Asia, as an ingredient for several kinds of industries considering their effectiveness, availability, and low production costs [2,3]. *Sargassum* spp. are the most prevalent algae species among brown algae belonging to the Phaeophyceae class, which inhabit tropical seawater ecosystems [4]. *Sargassum siliquastrum* is one of the edible brown algae extensively found in the Yellow Sea from the Republic of Korea to Japan [4]. Prior studies have identified that *S. siliquastrum* contains a considerable amount of fucoidan and has a wide range of potential applications, which made it a promising valuable marine resource [5].

Fucoidan is a water-soluble polysaccharide present in the cell walls of brown algae consisting of sulfated fucose units and trace amounts of other monosaccharides including galactose, mannose, glucose, arabinose, and xylose [6,7]. Studies have reported numerous biological activities of fucoidans isolated from different brown algae including anti-inflammatory [8], antioxidant [9], anticancer [6], antitumor [10], antidiabetic [11], and anti-allergic effects [12]. The efficacy of various biological activities of fucoidans varies with the source, structure, molecular weight, composition, and purity level [8]. As documented, low molecular weight fucoidan (LMF) has exhibited promising bioactivities and interestingly, it contained a relatively high degree of sulfation [13]. LMF with higher sulfate contents has been reported as an effective ROS inhibitor due to its high antioxidant capacity [5,9,13]. The antioxidant activity of LMF refined from *Sargassum horneri* isolated by following the step gradient ethanol precipitation against UVB-induced oxidative stress in human keratinocytes was investigated [9]. A prior in vivo study showed that type 2 diabetic mice treated with LMF derived from *Saccharina japonica* have the potential to reduce cerebrovascular damage by promoting angiogenesis [14]. Additionally, several investigations suggested that fucoidans from different brown algae have anti-inflammatory properties both in vitro and in vivo. One study has confirmed the potential anti-inflammatory activity of fucoidan from *Chnoospora minima* via inhibiting lipopolysaccharide (LPS) stimulated inflammatory responses in RAW 264.7 macrophages. Moreover, the results showed that the effect of LMF occurred by reducing the production of nitric oxide (NO), intracellular reactive oxygen species (ROS), and the expression of inducible nitric oxide synthase (iNOS) and cyclooxygenase (COX)-2 in zebrafish embryos stimulated with LPS. In another study, the LMF from *S. horneri* indicated protective effects via regulating inflammatory reactions and disrupted skin barrier functions in fine-dust-stimulated keratinocytes [13]. One of our previous studies found that LMF refined by *S. confusum* on FD-stimulated HaCaT keratinocytes recovered the cells from inflammatory reactions and protected underneath dermal fibroblasts from inflammation and extracellular matrix degradation [15].

Inflammation is a complicated biological reaction to injury, infection, or tissue damage involving various immune cells, chemical messengers, and other inflammatory mediators. However, exaggerated inflammatory responses can contribute to the development of several chronic illnesses including asthma, cardiovascular diseases, osteoporosis, cancer, obesity, and bronchitis [16]. With external stimuli, immune cells such as macrophages play a crucial role in inflammatory responses via releasing inflammatory mediators such as NO, prostaglandins, iNOS, COX-2, and inflammatory cytokines [13]. The activation of iNOS and COX-2, which affect platelet aggregation, vascular permeability, and thrombus formation, contribute to the abnormal release of NO and prostaglandin E_2_ (PGE_2_) [17]. The nuclear factor-kappa B (NF-κB) and mitogen-activated protein kinase (MAPK) signaling pathways are activated in stimulated cells, which regulate the expression of iNOS, COX-2, and the secretions of pro-inflammatory cytokines, such as tumor necrosis factor (TNF)-α, interleukin (IL)-1β and IL-6 [18]. In addition, the activation of the nucleotide-binding oligomerization domain- (NOD-) like receptor family pyrin domain-containing 3 (NLRP3) inflammasome multiprotein complex enhances inflammatory responses by regulating the release of pro-inflammatory cytokines; IL-18 and IL-1β in stimulated macrophages [19].

The LPS is a major component of gram-negative bacteria’s outer cell membrane that can trigger host inflammatory responses by increasing the production of cytokines, chemokines, and other pro-inflammatory mediators [20]. When macrophages are exposed to LPS during infections caused by microorganisms, they initiate the production of inflammatory mediators, subsequently leading to inflammatory responses [21]. Hence, inhibiting macrophage stimulations is a key therapeutic strategy for treating inflammatory diseases. One of our companion studies has found that LMF isolated from *S. siliquastrum* (FSSQ) indicates significant photoprotective activities while containing the highest sulfate amount in contrast to the high molecular weight fucoidans precipitated at the early stages [5]. Even though high sulfate content has an impact on the physical properties of polysaccharides, the relationship between sulfate content and bioactivities has yet to be clarified [22]. With the evaluation of the present understandings, we conclude that there is no evidence of the protective effect of FSSQ on inflammatory responses in vitro. This study was designed by hypothesizing that FSSQ suppresses the inflammatory responses by regulating the expression of inflammatory mediators in LPS-stimulated RAW 264.7 macrophages.

## 2. Results

### 2.1. FSSQ Effectively Increases Cell Viability and Suppresses the Intracellular ROS Production in LPS-Stimulated RAW 264.7 Macrophages

At the initial investigations with fucoidans isolated from *S. siliquastrum,* FSSQ indicated the best cytoprotective effect on LPS-stimulated RAW 264.7 macrophages (Appendix A). The cytotoxicity of the FSSQ treatment was determined by using concentrations ranging from 12.5 μg/mL to 200 μg/mL as a preliminary investigation. The 3-(4,5-dimethylthiazol-2-yl)-2,5-diphenyltetrazolium bromide (MTT) analysis indicated that the tested FSSQ concentrations up to 100 μg/mL had no significant cytotoxic effect on RAW 264.7 macrophages, as shown in Figure 1A. Based on the results, cell viability of RAW 264.7 macrophages decreased with LPS stimulation in contrast to the control group and considerably increased with FSSQ treatment in a dose-dependent manner up to 100 μg/mL (Figure 1B). Therefore, the concentrations of 25 μg/mL, 50 μg/mL, and 100 μg/mL were selected for the investigation. Furthermore, whether FSSQ treatment could have any effect on ROS accumulation in LPS-stimulated RAW 264.7 macrophages was investigated using 2′,7′-Dichlorofluorescin diacetate (DCF-DA) assays. According to the DCF-DA fluorometric results presented in Figure 1C, intracellular ROS production in LPS-stimulated RAW 264.7 macrophages promptly increased, while treatment of FSSQ dose-dependently reduced. A DCF-DA fluorescence microscopy analysis was conducted to confirm the ROS inhibitory effect of FSSQ against LPS-stimulated oxidative stress in RAW 264.7 macrophages. The resulting images in Figure 1D illustrated that LPS-stimulated RAW 264.7 macrophages had an increased ROS level, as shown by a high intensity of green fluorescence, in comparison to the control group. FSSQ pretreated RAW 264.7 macrophages and indicated the dose-dependent reduction in green fluorescence intensity. These results were strengthened by a DCF-DA flow cytometric analysis of LPS-stimulated RAW 264.7 macrophages with dose-dependent FSSQ treatment (Figure 1E). Indomethacin (IM) was used throughout the study as a positive control.

### 2.2. FSSQ Downregulates the Production of NO and PGE_2_, and the Expression of iNOS/COX-2

The effect of FSSQ treatment on NO and PGE_2_ production in LPS-stimulated RAW 264.7 macrophages was investigated using the Griess assay and the enzyme-linked immunosorbent assay (ELISA), respectively, to determine the anti-inflammatory effect of FSSQ. As shown in Figure 2A,B, the production of NO and PGE_2_ induced with LPS stimulation compared to the control group was significantly suppressed by FSSQ treatment in a dose-dependent manner. Subsequently, the interaction of the iNOS and COX-2 inhibitions and the inhibitory effect of FSSQ on NO and PGE_2_ production was examined. According to the Western blot analysis, the protein expression of iNOS and COX-2 increased with LPS stimulation, whereas FSSQ treatment dose-dependently downregulated iNOS and COX-2 expression (Figure 2C).

### 2.3. FSSQ Suppresses the Expression of Inflammatory Cytokines

A reverse transcription polymerase chain reaction (RT-PCR) analysis was used to measure the effect of FSSQ on mRNA expression levels of IL-1β, IL-6, and TNF-α in LPS-stimulated RAW 264.7 macrophages. As illustrated in Figure 3A, mRNA expression levels of IL-1β, IL-6, and TNF-α were markedly increased following LPS stimulation, while pretreatment with FSSQ considerably attenuated the mRNA expression levels of the inflammatory cytokines in LPS-stimulated RAW 264.7 macrophages in a dose-dependent manner. An ELISA analysis was conducted to confirm the anti-inflammatory potential of FSSQ treatment. According to the findings of the ELISA analysis, FSSQ treatment dose-dependently reduced the production of IL-1β, IL-6, and TNF-α in LPS-stimulated RAW 264.7 macrophage cell culture media (Figure 3B).

### 2.4. FSSQ Suppresses the Activation of NF-κB and MAPK Signaling Pathways

The NF-κB and MAPK signaling pathways are essential regulators of inflammation in response to various stimuli. Based on a Western blot analysis, the phosphorylation level of MAPK signaling pathways, including p38, ERK, and JNK, was considerably increased in LPS-stimulated RAW 264.7 macrophages, while dose-dependently decreased by FSSQ treatment (Figure 4A). Additionally, activation of NF-κB signaling is triggered by LPS stimulation, leading to phosphorylation and allowing translocation of NF-κB into the nucleus. Figure 4B illustrated that phosphorylation of cytosolic IκBα and NF-κB p65 increased by LPS stimulation and significantly decreased by FSSQ treatment in a dose-dependent manner. Similarly, FSSQ treatment dose-dependently downregulated nuclear translocation of NF-κB p65 in LPS-stimulated RAW 264.7 macrophages (Figure 4C). In addition, nuclear translocation of NF-κB p65 was assessed by immunofluorescence analysis (Figure 4D). The detection of NF-κB p65 nuclear translocation can be analyzed by investigating the green fluorescence level in the nucleus. The outcomes of the immunofluorescence analysis further revealed that LPS stimulation increases NF-κB p65 nuclear translocation compared to the control. However, FSSQ treatment decreased the NF-κB p65 nuclear translocation in LPS-stimulated RAW 264.7 macrophages dose-dependently.

### 2.5. FSSQ Decreases the Expression of NLRP3 Inflammasome Molecules

The component proteins of NLRP3 inflammasome, including NLPR3, caspase-1, and apoptosis-associated speck-like protein containing CARD domain (ASC), play a crucial role in the immune response by regulating the release of pro-inflammatory cytokines, such as IL-1β and IL-18 [23]. In the present study, the effect of FSSQ on the activation of the component proteins involved in inflammasome and the secretion of IL-1β and IL-18 in LPS-stimulated RAW 264.7 macrophages was observed by a Western blot analysis. As indicated in Figure 5, protein expression of NLPR3, ASC, caspase-1, IL-18, and IL-1β was increased following LPS stimulation compared to the control group. However, FSSQ significantly downregulated the protein expression of NLPR3, ASC, caspase-1, IL-18, and IL-1β in a dose-dependent manner. These results suggest the potential for FSSQ to reduce inflammation by inhibiting the expression of NLRP3 inflammasome molecules and the subsequent release of pro-inflammatory cytokines.

### 2.6. FSSQ Upregulates the Activation of Nrf2/HO-1 Signaling Pathway

The Nrf2/HO-1 signaling pathway is an important part of the cellular response to oxidative stress and inflammation [24]. Figure 6A showed that LPS stimulation reduced the expression of the cytosolic HO-1, NQO1, and nuclear-translocated Nrf2 proteins in RAW 264.7 macrophages, whereas the dose-dependent effects of FSSQ treatment significantly increased these protein expression levels. Additionally, by downregulating HO-1 activity in the presence of zinc protoporphyrin IX (ZnPP), an HO-1 inhibitor, the cytoprotective effect of FSSQ on cell viability, NO generation, and intracellular ROS production in LPS-stimulated RAW 264.7 macrophages were investigated. When LPS stimulation increased NO and ROS production while decreasing cell viability, FSSQ treatment significantly increased cell viability and decreased NO and ROS production at a dose of 100 µg/mL. The ZnPP significantly reduced the increase in cell viability while increasing NO and ROS generation, which was reduced by FSSQ treatment (Figure 6B–D).

## 3. Discussion

According to previous studies, the brown alga *S. siliquastrum* has been shown to possess a variety of bioactive metabolites with numerous bioactivities such as sulfated polysaccharides, phlorotannins, fucoxanthin, sargachromanols, and sargaquinoic acid [5,25,26,27]. One of the previous studies revealed the anti-inflammatory effect of sargaquinoic acid isolated from *S. siliquastrum* on NO production in LPS-induced macrophages via inhibiting NF-κB and c-Jun *N*-terminal kinase pathways [27]. In another study, the antioxidant effect of LMF from *S. siliquastrum* against oxidative stress in UVB-exposed keratinocytes was investigated [5]. Furthermore, the action mechanism of sargachromanol G obtained from *S. siliquastrum* on inflammatory responses has been examined in LPS-stimulated RAW 264.7 macrophages [28]. Therefore, a range of experiments was conducted to investigate the effect of FSSQ against inflammatory reactions in LPS-stimulated RAW 264.7 macrophages in the present study.

RAW 264.7 macrophages activated by LPS stimulation led to the production of inflammatory mediators including NO, PGE_2_, and inflammatory cytokines, and the activation of the various signaling pathways related to the inflammatory responses. The synthesis of NO from L-arginine is catalyzed by NO synthase. The NO is essential for regulating physiological functions such as neurotransmission, vasodilation, and immunological defense [29]. LPS-stimulated RAW 264.7 macrophages promote excessive NO production as a response to inflammatory stimuli and trigger the accumulation of intracellular ROS, which causes oxidative stress and cell damage. In addition, ROS is actively involved in regulating the NF-κB signaling cascade [30]. Thus, the downregulation of NO and ROS production could affect the reduction in inflammatory responses. The findings of the study showed that intracellular ROS and NO production significantly decreased with FSSQ treatment in a dose-dependent manner. The presence of iNOS and COX-2 in cytosol involves the production of NO and PGE_2_ in macrophages. The iNOS is the enzyme responsible for the production of NO. The COX enzyme catalyzes the arachidonic acid pathway, which contributes to cellular PGE_2_ production. The major COX enzyme, COX-2, regulates the generation of PGE_2_ during inflammatory reactions in macrophages [30]. Furthermore, the expression of iNOS and COX-2 are correlated with the expression of pro-inflammatory cytokines in LPS-stimulated RAW 264.7 macrophages [29]. The present study performed a dose-dependent inhibition of PGE_2_ production, iNOS expression, and COX-2 expression, indicating an anti-inflammatory effect of FSSQ treatment in stimulated RAW 264.7 macrophages. As reported, the production of pro-inflammatory cytokines IL-1β, IL-6, and TNF-α increased in LPS-stimulated RAW 264.7 macrophages, while the anti-inflammatory cytokine IL-10 is considered a crucial target for anti-inflammatory effects [30,31,32]. TNF-α and IL-1β bare a major role in a variety of acute and chronic inflammatory illnesses by acting as inducers of endothelial adhesion molecules that are implicated in the inflammatory response [33]. IL-6 and IL-1 in human synovial cells synergistically involve inflammatory conditions [34]. Several studies have indicated that sulfated polysaccharides from brown seaweed possess the potential to downregulate the expression of pro-inflammatory cytokines in macrophages stimulated with LPS [25,30]. Interestingly, the results of the RT-PCR and ELISA analyses in the present study revealed that pre-treatment with FSSQ significantly reduced the expression of TNF-α, IL-1β, and IL-6 in LPS-stimulated RAW 264.7 macrophages. Therefore, the inhibition of inflammatory cytokines provides valuable insights into the development of effective products from FSSQ against inflammatory responses.

Furthermore, the activation of MAPK and NF-κB signaling pathways is known to have important roles in the regulation of inflammatory responses. Upon stimulation of 264.7 macrophages with LPS, toll-like receptor 4 (TLR4) on the cell membrane recognizes and specifically binds to LPS, leading to the recruitment of adaptor proteins. The common adaptor for the activation of the TLR pathway has been identified as the MyD88 component, and these adaptor proteins then activate downstream signaling cascades, activating the MAPK and NF-B pathways [35]. The MAPK and NF-κB pathways are interconnected, and the MAPK superfamily consists of three subgroups, including p38, JNK, and ERK. Previous reports have indicated that the phosphorylation of p38, JNK, and ERK was dramatically elevated with LPS stimulation in macrophages, leading to inflammation [36]. However, treatment with FSSQ resulted in a dose-dependent modulation of MAPK phosphorylation in LPS-stimulated RAW 264.7 macrophages. NF-κB is an essential regulator of transcription involved in inflammation, cell proliferation, immunity, and apoptosis [37]. LPS stimulation of the NF-κB pathway results in the phosphorylation and degradation of cytosolic IκB proteins by the IκB kinase complex in the cytoplasm, followed by nuclear translocation of NF-kB p65 [35]. Nuclear translocation of NF-κB p65 triggers the transcription of genes encoding inflammatory mediators, such as iNOS and COX-2, as well as pro-inflammatory cytokines including IL-1β, IL-6, and TNF-α [33,36]. The pathogenesis of cancer and chronic inflammatory illnesses is also greatly influenced by NF-κB phosphorylation. Therefore, inhibiting the expression of proteins associated with the NF-κB signaling cascade has been identified as a promising therapeutic target for inflammatory disorders. Interestingly, our results indicated that FSSQ suppressed the activation of NF-κB signaling via inhibiting the phosphorylation of NF-κB molecules as well as NF-κB p65 nuclear translocation in the LPS-stimulated RAW 264.7 macrophages. Additionally, the NLRP3 inflammasome, which regulates pyroptosis, has a significant impact on the LPS-stimulated inflammatory conditions in the RAW 264.7 macrophages. The NLRP3 inflammasome multiprotein complex includes a sensor (NLRP3), an adaptor (ASC), and an effector (caspase-1) [19]. Upon activation, NLRP3 combines with ASC and subsequently cleavage procaspase-1 into its active form, caspase-1, which induces the maturation and release of pro-inflammatory cytokines, such as IL-1β and IL-18, leading to further amplification of the inflammatory response [38]. Furthermore, activation of the NLRP3 inflammasome is associated with insulin resistance and related diseases, including Alzheimer’s disease, obesity, and cardiovascular disease [39]. Therefore, several studies have suggested therapies with the ability to inhibit abnormal NLRP3 inflammasome activation [19,38,40]. In this study, FSSQ significantly attenuated the protein expression of NLPR3, ASC, and caspase-1, as well as the subsequent release of pro-inflammatory cytokines IL-1β and IL-18 in LPS-stimulated RAW 264.7 macrophages.

Excessive accumulation of intracellular ROS, which is considered the most potent inflammatory mediator, can cause oxidative stress, and trigger the initiation of inflammatory responses [8,41]. However, prior studies have shown that the Nrf2/HO-1 signaling pathway inhibits intracellular ROS and protects against inflammatory reactions in stimulated RAW 264.7 macrophages by upregulating the expression of the antioxidant gene [42,43]. Under unstimulating conditions, the cytosolic Nrf2 is bound to its negative regulator, actin-binding protein Keap1 (Kelch-like ECH-associated protein 1). During oxidative stress, Nrf2 dissociates from Keap 1 and translocates into the nucleus, where it binds to the antioxidant-response element (ARE) in the promoters of gene-encoding antioxidant enzymes, including NQO1 and HO-1 [8,42]. HO-1 is an important antioxidant, anti-inflammatory, and cytoprotective enzyme that is effectively involved in the inhibition of intracellular ROS production and subsequent inflammatory responses [41]. The results of the current study revealed that FSSQ considerably increased Nrf2 nuclear translocation and cytosolic NQO1 and HO-1 expression in LPS-stimulated RAW 264.7 macrophages in a dose-dependent manner, upregulating antioxidant enzyme activity. Additionally, the study confirmed the cytoprotective effect of FSSQ using ZnPP on cell viability, intracellular ROS production, and NO production in LPS-stimulated RAW 264.7 macrophages. According to the outcomes of the study, ZnPP reduced the cytoprotective effect of FSSQ treatment via inhibiting HO-1 activity and weakened the ROS and NO scavenging ability. The results demonstrated that FSSQ activated the Nrf2/HO-1 signaling pathway on LPS-stimulated oxidative stress and inflammatory responses in RAW264.7 macrophages. Considering industrial requirements and low production costs, mass purification of LMF having anti-inflammatory activities has the potential to develop and/or incorporate as a functional food, nutraceutical, or cosmeceutical. In addition, scientific data in this paper would be helpful to commercially grow *S. siliquastrum* as a functional material profitably.

## 4. Materials and Methods

### 4.1. Materials

The murine macrophage cell line RAW 264.7 was purchased from the Korean Cell Line Bank (KCLB, Seoul, Republic of Korea). Fetal bovine serum (FBS), penicillin/streptomycin, and Dulbecco’s Modified Eagle Medium (DMEM) were purchased from Gibco (Grand Island, NY, USA). Sigma-Aldrich (St. Louis, MO, USA) provided the MTT, DCF-DA, dimethyl sulfoxide (DMSO), IM, ZnPP, bovine serum albumin (BSA), agarose, paraformaldehyde, Triton^TM^ X-100, TRIzol, chloroform, isopropanol, and ethidium bromide. An Ace-α-^®^ cDNA synthesis kit was supplied by ReverTra (Toyobo, Osaka, Japan). A BCA protein assay kit, Pierce™ RIPA buffer, a NE-PER^®^ nuclear and cytoplasmic extraction kit, protein ladder, 1-Step transfer buffer, and diethylpyrocarbonate (DEPC) water were purchased from Thermo Fisher Scientific (Rockford, IL, USA). Skim milk powder was obtained from BD Difco™ (Sparks, MD, USA). Primary and secondary antibodies, Prolong^®^ Gold antifade reagent with DAPI, normal goat serum, and DyLightTM 554 Phalloidin were provided by Cell Signaling Technology (Beverly, MA, USA). The PCR primers were supplied by Bioneer Inc. (Deadeock-gu, Daejeon, Republic of Korea). ELISA kits for IL-1β, -6, and TNF-α were purchased from R & D System Inc. (Minneapolis, MN, USA).

### 4.2. Isolation and Purification Method of FSSQ from S. siliquastrum

FSSQ was obtained by following the method figured out by Fernando et al., 2020 [5]. The structural and compositional analysis of the FSSQ were mentioned in the previous study [5]. In brief, *S. siliquastrum* samples were collected from the Jeju coast (an island that belongs to the Republic of Korea), washed, air-dried, and pulverized into fine particles. Then, the powder was depigmented by washing with a 1:1 chloroform and methanol (*v*/*v*) mixture. Then, the powder was soaked in a 1:9 formaldehyde-ethanol solution (*v*/*v*) for 5 h with continuous stirring at 40 °C. Following the 80% ethanol washing, the powder was dried in a forced hot air-drying oven at 50 °C for 2 days. Fucoidans were obtained by celluclast-assisted enzymatic extraction. The resulting fucoidan was separated into groups following the step gradient ethanol precipitation method. FSSQ was obtained at the final precipitation in the series. As per the previous investigation, the polysaccharide and sulfate content of the FSSQ fraction were 45.96 ± 0.36% and 21.92 ± 0.46%, respectively. The molecular weight (MW) of the FSSQ fraction was estimated by agarose gel electrophoresis with MW markers. The estimated MWs were spread between 8 to 25 kDa approximately. As per the analysis, FSSQ indicated the highest sulfate content in contrast to the relatively high molecular weighted fucoidans precipitated at the early stages of the isolation. Further, fucoidan was identified by using FTIR analysis. Results indicated the prominent peaks relevant to the common polysaccharides and sulfate groups. FSSQ contained 72.67 % fucose, 2.35 % galactose, 3.73 % glucose, 9.26 % mannose, and 11.99 % other monosaccharides [5].

### 4.3. RAW Cell Culture, LPS Stimulation, and FSSQ Treatment

The RAW 264.7 macrophages were cultured in a humidified incubator with 5% CO_2_ at 37 °C. DMEM supplemented with 10% FBS and a 1% penicillin/streptomycin mixture was used to maintain the RAW 264.7 macrophages. Sub-cultures were performed once every two days until they achieved exponential growth that was suitable for in vitro experiments. Before being used in the experiments, FSSQ was diluted with PBS to create final treatment concentrations of 12.5, 25, 50, 100, and 200 μg/mL. Cells were stimulated with LPS at a concentration of 1 μg/mL throughout the study.

### 4.4. Cell Viability Assay

Following the method of the prior study, the effect of FSSQ on cell viability was evaluated using the MTT assay [8]. In brief, 1 × 10^4^ cells/well were seeded in 96-well plates and incubated for 24 h in the conditions mentioned in Section 4.3. Before being stimulated with LPS, cells were pretreated with various concentrations of FSSQ, ranging from 12.5 μg/mL to 200 μg/mL, for 1 h. The 15 µL of the MTT reagent (5 mg/mL) was added to each well after 24 h of stimulation and then incubated for 4 h at 37 °C. The created formazan crystals in wells were dissolved in 100 µL of DMSO for 30 min after the media were removed. Finally, a SpectraMax M2 microplate reader was used to measure absorbance at 570 nm (Molecular Devices, Silicon Valley, CA, USA).

### 4.5. Investigation of Intracellular ROS Production

The protective effect of FSSQ against intracellular ROS production in LPS-stimulated RAW 264.7 macrophages was investigated by using DCF-DA assay [15]. Following a 24 h incubation period, cells were treated with FSSQ sample doses and incubated for 1 h. Then cells were stimulated with LPS. The DCF-DA solution was added to the cells after 1 h of cell stimulation, and intracellular ROS generation in the cells was measured using a SpectraMax M2 microplate reader at 485 and 528 nm for excitation and emission, respectively. To further confirm the protective effect of FSSQ against oxidative stress, the DCF-DA-treated cells were also investigated using a Thermo Fisher Scientific EVOS M5000 Imaging fluorescence microscope (Rockford, IL, USA) and a CytoFLEX flow cytometer (Beckman Coulter, Brea, CA, USA).

### 4.6. Measurement of NO Production

The NO production in LPS-stimulated RAW 264.7 macrophages was conducted using the Griess assay. The cells at the 2 × 10^4^ cells/well density were pretreated with FSSQ sample concentrations before being stimulated with LPS for 24 h. Subsequently, a Griess reagent (1% sulfanilamide and 0.1% N-[naphthyl] ethylenediamine dihydrochloride in 2.5% H_3_PO_4_) was mixed with a cell supernatant from each well for 10 min at room temperature (20–22 °C). A SpectraMax M2 microplate reader was used to measure the absorbance at 540 nm.

### 4.7. Western Blot Analysis

A Western blot analysis was carried out using the same procedure as the earlier study [44]. FSSQ pretreated, LPS-stimulated RAW 264.7 macrophages were harvested and lysed to isolate cytoplasmic and nuclear proteins using the cytoplasmic and NE-PER^®^ nuclear extraction kits. After determining the protein concentrations, 35 µg of protein from each lysate was subjected to electrophoresis on 10% polyacrylamide gels. Transferred protein bands onto nitrocellulose membranes were then incubated with monoclonal primary antibodies (1:1000) and HRP-conjugated secondary antibodies (1:3000) after being blocked with 5% skim milk in TBST. Enhanced chemiluminescence (ECL) reagents (Cyanagen Srl, Bologna, Italy) were used to visualize the acquired bands on a Core Bio Davinch-ChemiTM imaging system (Seoul, Republic of Korea).

### 4.8. RNA Extraction and RT-PCR Analysis

The mRNA expression of inflammatory molecules was measured using an RT-PCR analysis following the method described by Kirindage et al. [45]. Briefly, the total RNA was isolated from the cells using TRIzol, chloroform, and isopropanol reagents. A ReverTra Ace-α-^®^ cDNA synthesis kit was used to synthesize cDNA from total RNA following the manufacturer’s instructions using a TaKaRa PCR Thermal Cycler (TaKaRa Bio Inc., Otsu, Japan). Following electrophoresis on 1.5% agarose gels with 0.5 µg/mL ethidium bromide, the RT-PCR products were then visualized using a Wisd WUV-L20 UV transilluminator (Daihan Scientific Co., Wonju-si, Republic of Korea). The relative intensities of expression levels were quantified using ImageJ software (Version 1.52a, US National Institutes of Health, Bethesda, MD, USA).

### 4.9. ELISA Analysis

The cell-cultured media in the multi-well plates of all RAW 264.7 macrophage groups were collected. Subsequently, following the manufacturer’s instructions, the production of inflammatory cytokines in the collected cell-cultured media was analyzed using the relevant ELISA assay kits. Absorbance at 450 nm was measured using a SpectraMax M2 microplate reader. Standard curves were used to normalize the samples.

### 4.10. Immunofluorescence Assay

The RAW 264.7 macrophages (1 × 10^4^ cells/chamber) were cultured in chamber slides and incubated for 24 h before sample treatment. After 30 min of LPS stimulation, wells were washed with PBS and fixed with 4% formaldehyde. Before being incubated with primary antibodies (anti-NF-κB p65) overnight, cells were blocked using a blocking buffer (PBS containing 5% normal goat serum and 0.3% Triton^TM^ X-100) for 1 h. The cells were then treated for 2 h with Alexa Fluor^®^ 488 conjugated anti-mouse IgG. The slides were covered with coverslips using Prolong^®^ Gold antifade reagent with DAPI after PBS washing, and cells were visualized using an EVOS M5000 Imaging fluorescence microscope from Thermo Fisher Scientific.

### 4.11. Statistical Analysis

The data were expressed as mean ± standard error (SE), and experiments were carried out in triplicate (*n* = 3). Statistical comparisons were conducted using one-way analysis of variance (ANOVA) followed by Duncan’s multiple range test using IBM SPSS software (Version 24.0, Chicago, IL, USA), and statistical significance was defined as *p* < 0.05.

## 5. Conclusions

In summary, the results suggest that FSSQ significantly downregulated the expression of inflammatory mediators such as NO, PGE_2_, iNOS, COX-2, and pro-inflammatory cytokines via regulating the MAPK and NF-κB signaling pathways in LPS-stimulated RAW 264.7 macrophages. FSSQ suppressed NLRP3 inflammasome molecules expression in stimulated RAW 264.7 macrophages. Moreover, a Western blot analysis indicated the cytoprotective effect of the FSSQ relay on Nrf2/HO-1 signaling. The findings revealed the potent anti-inflammatory effect of FSSQ in vitro.

## Figures and Tables

**Figure 1 marinedrugs-21-00347-f001:**
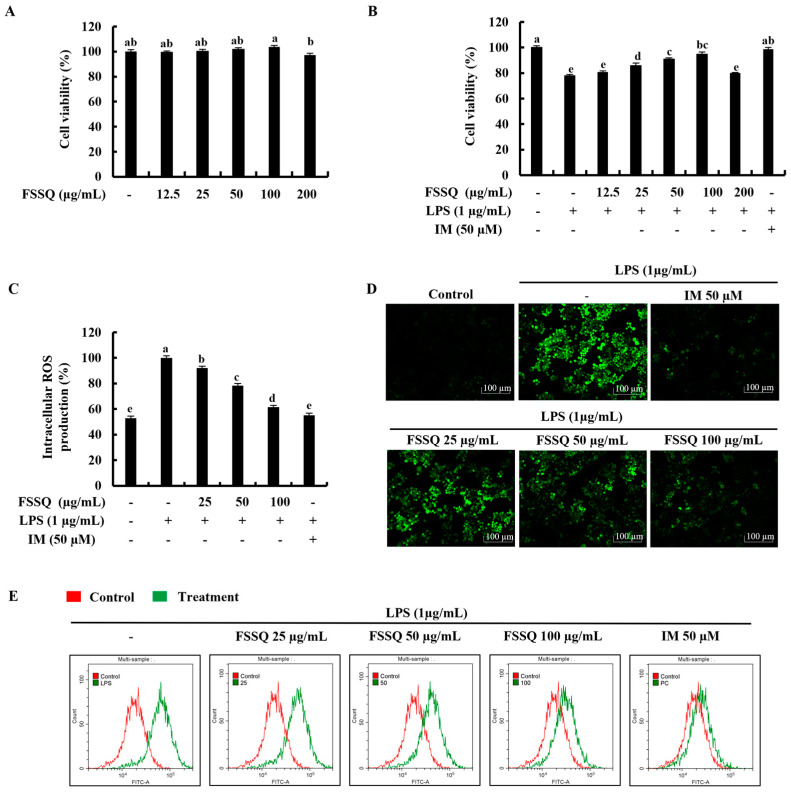
Protective effect of FSSQ in LPS-stimulated RAW 264.7 macrophages. (**A**) Dose-dependent cytotoxicity and (**B**) cell viability. Investigation of intracellular ROS production by (**C**) fluorometric analysis, (**D**) fluorescence microscopy, and (**E**) flow cytometry, with and without LPS stimulation. The values represent data from three distinct experiments (n = 3) and are shown as the mean ± standard error (SE). Significantly different results were indicated with different lowercase English letters (*p* ˂ 0.05).

**Figure 2 marinedrugs-21-00347-f002:**
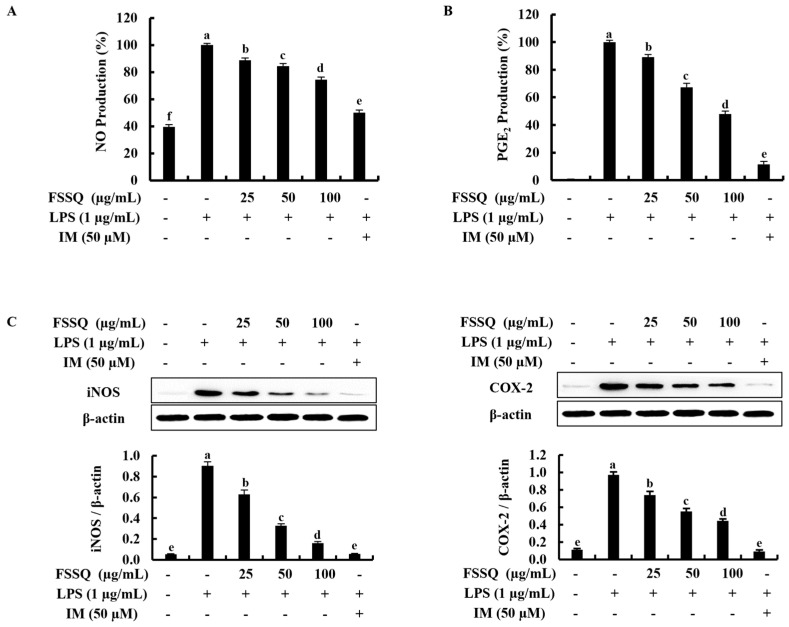
The protective effect of FSSQ on the production of inflammatory regulators in LPS-stimulated RAW 264.7 macrophages. The effect of FSSQ on (**A**) NO production and (**B**) PGE_2_ production in stimulated RAW 264.7 macrophages. A Western blot analysis of (**C**) iNOS and COX-2 protein expression in LPS-stimulated RAW 264.7 macrophages. The values represent data from three distinct experiments (n = 3) and are shown as the mean ± standard error (SE). Significantly different results were indicated with different lowercase English letters (*p* ˂ 0.05).

**Figure 3 marinedrugs-21-00347-f003:**
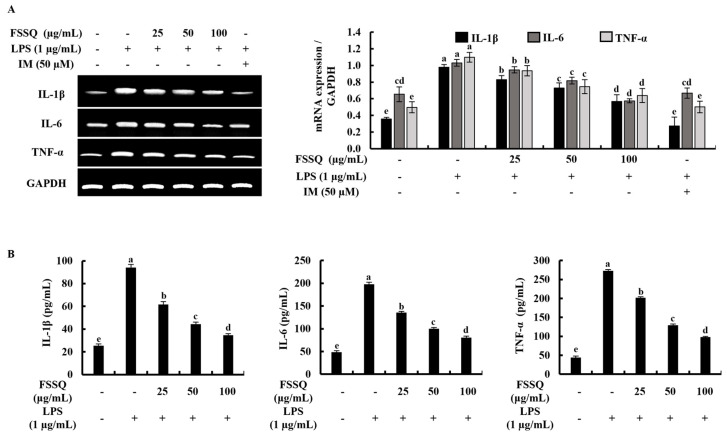
Inhibitory effect of FSSQ on inflammatory cytokines expression in LPS-stimulated RAW 264.7 macrophages. (**A**) An RT-PCR analysis evaluated the mRNA expression of inflammatory cytokines in stimulated RAW 264.7 macrophages. The effect of FSSQ on (**B**) IL-1β, IL-6, and TNF-α production in LPS-stimulated cell culture media was analyzed using ELISA kits. The values represent data from three distinct experiments (n = 3) and are shown as the mean ± standard error (SE). Significantly different results were indicated with different lowercase English letters (*p* ˂ 0.05).

**Figure 4 marinedrugs-21-00347-f004:**
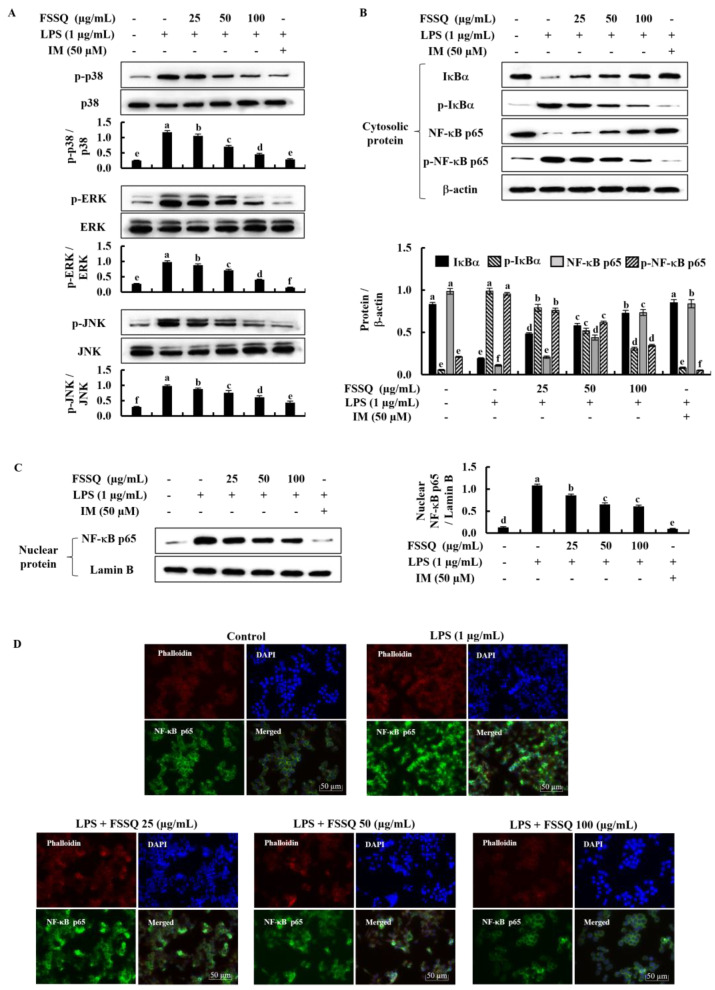
The anti-inflammatory effect of FSSQ on MAPK and NF-κB molecular mediators in LPS-stimulated RAW 264.7 macrophages. A Western blot analysis of (**A**) MAPK, (**B**) cytosolic NF-κB protein expression, and (**C**) nucleic NF-κB p65 protein levels in stimulated RAW 264.7 macrophages affected by FSSQ treatment. (**D**) An immunofluorescence analysis of NF-κB p65 nuclear translocation. The values represent data from three distinct experiments (n = 3) and are shown as the mean ± standard error (SE). Significantly different results were indicated with different lowercase English letters (*p* ˂ 0.05).

**Figure 5 marinedrugs-21-00347-f005:**
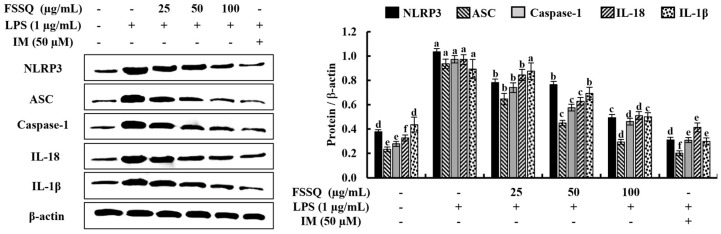
The protective effect of FSSQ on the expression of NLRP3 inflammasome molecules in LPS-stimulated RAW 264.7 macrophages. The values represent data from three distinct experiments (n = 3) and are shown as the mean ± standard error (SE). Significantly different results were indicated with different lowercase English letters (*p* ˂ 0.05).

**Figure 6 marinedrugs-21-00347-f006:**
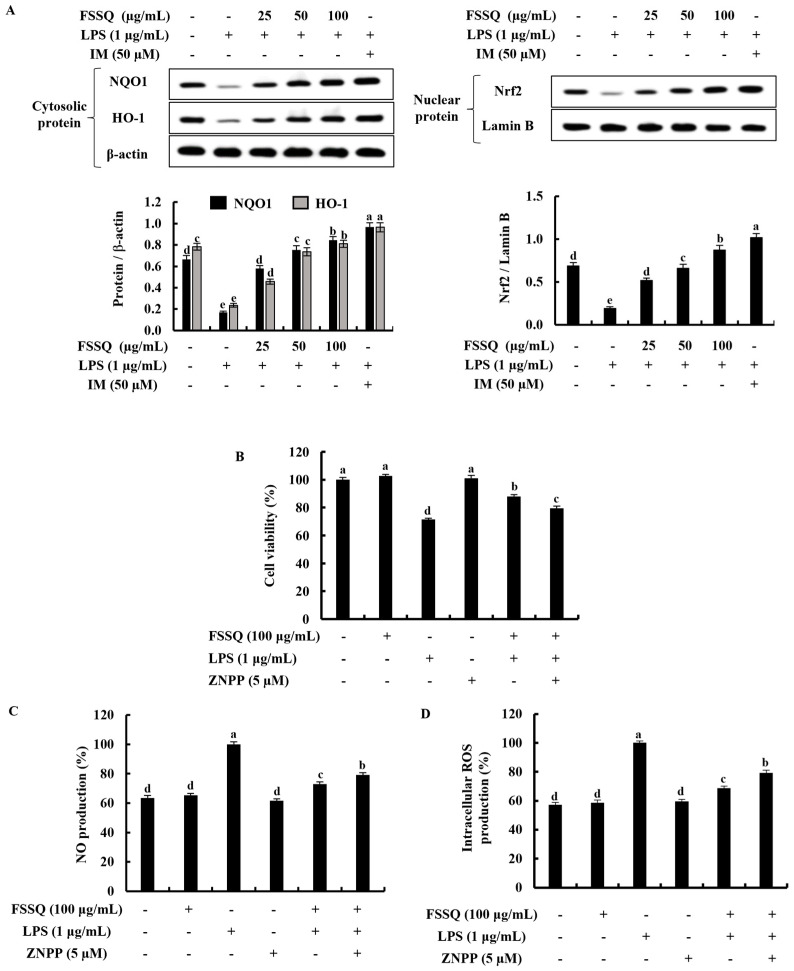
The cytoprotective effect of FSSQ on activation of the Nrf2/HO-1 signaling in LPS-stimulated RAW 264.7 macrophages. Analysis of (**A**) dose-dependent response of FSSQ on cytosolic HO-1, NQO1, and nuclear translocated Nrf2 protein expression by a Western blot analysis. The effect of ZnPP on FSSQ treated (**B**) cell viability, (**C**) NO production, and (**D**) intracellular ROS production in LPS-stimulated RAW 264.7 macrophages. The values represent data from three distinct experiments (n = 3) and are shown as the mean ± standard error (SE). Significantly different results were indicated with different lowercase English letters (*p* ˂ 0.05).

## Data Availability

The data presented in this study are available upon request from the corresponding author.

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
