# Peer review of "The Anti-Inflammatory Effect of Low Molecular Weight Fucoidan from Sargassum siliquastrum in Lipopolysaccharide-Stimulated RAW 264.7 Macrophages via Inhibiting NF-κB/MAPK Signaling Pathways"

_marinedrugs, 2023, doi:10.3390/md21060347_

Round 1

Reviewer 1 Report

In this manuscript of " Anti-Inflammatory Effect of Low Molecular Weight Fucoidan from Sargassum siliquastrum in Lipopolysaccharide-Stimulated RAW 264.7 Macrophages via Inhibiting NF-κB/MAPK signaling pathways", it fully demonstrates the authors' accumulation of knowledge in the field and suggests new trends for other scholars to follow. But this needs revision and the author reply comments properly before accepted. The comments and questions are as follows:

 1.     What is the impact of the amount of sulfate and molecular weight of fucoidan on its activities?

2.     How did FSSQ inhibit the production of NO and prostaglandin E2 in LPS-stimulated RAW 264.7 macrophages?

3.     In the introduction part, it is recommended that more convincing literature be added, e.g. 10.3390/foods12040878.

4.     What happens to the cytoprotective effect of FSSQ when HO-1 activity is suppressed by ZnPP?

5.     What implications did these findings have for the potential use of FSSQ against inflammatory responses in LPS-stimulated RAW 264.7 macrophages?

Above all, I suggest this can be accepted before making revision or answer the questions properly.

Reviewer 2 Report

The current research addresses the anti-inflammatory effect of low molecular weight fucoidan isolated from the brown alga Sargassum siliquastrum on lipopolysaccharide (LPS)-stimulated inflammatory responses in RAW 264.7 macrophages. However, the anti-inflammatory effect of fucoidan via inhibition of NF-κB/MAPK and NLRP3 Inflammasome activation has been discussed and reviewed extensively in previous reports. Then, I cannot determine the work's novelty, even if the fucoidan's source is novel from Sargassum siliquastrum.

The authors presented previously a related work in International Journal of Biological Macromolecules (Step gradient alcohol precipitation for the purification of low molecular weight fucoidan from Sargassum siliquastrum and its UVB protective effects). It has been understood that SSQC4 fraction was used in the current study. Then, a number of drawbacks should be clarified:

-          The extraction method was based on ethanol precipitation which results in crude fucoidan production. The chemical analysis shows the presence of other impurities such as polyphenols and proteins.

-          It was recommended to perform a comparative study between the different fractions to show which one is the most effective and then conduct further investigations! Finally, we may conclude that the low molecular fraction was the best.

-          The title starts with low molecular and then the abstract pointed out that the sulfate content also contributes. May the sulfate content contribute more than the molecular weight? Especially that this fraction showed about 21% sulfate more than the other fraction.

-          Other bioactive compounds beside to fucoidan may also interfere with the biological evaluation such as phlorotannins, fucoxanthin, sargachromanols, and sargaquinoic acid. The absence of these constituents should be confirmed.

-          The histopathology, including histo-immunoassays, part is missing.

Conclusively, there is debate regarding the choice of this fraction. A more comprehensive study based on structure-activity relationships is recommended. 

It is fine and understandable. Only fine editing is just required.

Round 2

Reviewer 2 Report

Thanks for the authors improving the quality of the manuscript. I would recommend to add the significance levels between the investigated doses in Fig. S1. The manuscript may be accepted after this minor revision.

The English is fine.
